# A kidney-hypothalamus axis promotes compensatory glucose production in response to glycosuria

Tumininu S Faniyan[1], Xinyi Zhang[2], Donald A Morgan[3], Jorge Robles[1], Siresha Bathina[1], Paul S Brookes[4], Kamal Rahmouni[3], Rachel J Perry[2], Kavaljit H Chhabra[1]*

[1]Department of Medicine, Division of Endocrinology, Diabetes, and Metabolism, University of Rochester Medical Center, Rochester, United States; [2]Department of Cellular and Molecular Physiology, Yale University, New Haven, United States; [3]Department of Neuroscience and Pharmacology, University of Iowa Carver College of Medicine, Iowa City, United States; [4]Department of Anesthesiology, University of Rochester Medical Center, Rochester, United States

*For correspondence: kavaljit_chhabra@UKY.Edu

Competing interest: The authors declare that no competing interests exist.

**Abstract** The kidneys facilitate energy conservation through reabsorption of nutrients including glucose. Almost all the filtered blood glucose is reabsorbed by the kidneys. Loss of glucose in urine (glycosuria) is offset by an increase in endogenous glucose production to maintain normal energy supply in the body. How the body senses this glucose loss and consequently enhances glucose production is unclear. Using renal *Slc2a2* (also known as *Glut2*) knockout mice, we demonstrate that elevated glycosuria activates the hypothalamic-pituitary-adrenal axis, which in turn drives endogenous glucose production. This phenotype was attenuated by selective afferent renal denervation, indicating the involvement of the afferent nerves in promoting the compensatory increase in glucose production. In addition, through plasma proteomics analyses we observed that acute phase proteins - which are usually involved in the body's defense mechanisms against a threat – were the top candidates which were either upregulated or downregulated in renal *Slc2a2* KO mice. Overall, afferent renal nerves contribute to promoting endogenous glucose production in response to elevated glycosuria and loss of glucose in urine is sensed as a biological threat in mice. These findings may be useful in improving the efficiency of drugs like SGLT2 inhibitors that are intended to treat hyperglycemia by enhancing glycosuria but are met with a compensatory increase in endogenous glucose production.

## eLife assessment

The study presents **valuable** findings on compensatory mechanisms in response to glycosuria. The evidence supporting the claims is **solid**, although a causal relationship is somewhat uncertain and the addition of a more clinically relevant model would have strengthened the findings. The work will be of interest to diabetes investigators.

## Introduction

The kidneys help conserve energy by reabsorbing nutrients and preventing their loss in urine. Almost all the filtered glucose is reabsorbed by the kidneys. The renal threshold for glucose reabsorption in humans is 180 mg/dl and in rodents is about 400 mg/dl (*Noonan and Banks, 2000*). In conditions such as overt hyperglycemia (beyond the renal glucose threshold), genetic loss of glucose transporters like

GLUT2 (*de Souza Cordeiro et al., 2021*; *de Souza Cordeiro et al., 2022*; *Sakamoto et al., 2000*) or SGLT2 (*Kleta et al., 2004*), and/or renal dysfunction, glucose is excreted in the urine. Enhancing glycosuria, through SGLT2 inhibition (*Rossetti et al., 1987*; *Oku et al., 1999*; *Han et al., 2008*), is now used to reduce blood glucose levels in humans with diabetes. However, the glucose loss is offset by a compensatory increase in endogenous glucose production, consequently compromising the efficacy of this strategy in the treatment of diabetes (*Solis-Herrera et al., 2020*; *Liu et al., 2012*). We recently produced renal *Slc2a2* (also known as *Glut2*) knockout mice (*de Souza Cordeiro et al., 2022*), which show massive glycosuria and are protected from diabetes and diet-induced obesity. Despite the loss of glucose in the urine, renal *Slc2a2* knockout mice maintain normal fasting blood glucose levels. Collectively, these observations indicate the presence of a fundamental mechanism involved in sensing glucose loss and triggering a compensatory increase in endogenous glucose production in rodents and humans. This mechanism is also aligned with the established survival strategies that reduce utilization and increase the production of glucose (energy) during a flight or fight response against a threat (*Surwit et al., 1992*; *Marik and Bellomo, 2013*).

Understanding how the kidney senses glucose loss and consequently activates compensatory pathways to make up for this loss will enhance our knowledge about integrative mechanisms that regulate glucose homeostasis. Answering such research questions may also be useful in medicine to increase efficiency of drugs such as SGLT2 inhibitors that are used to reduce blood glucose levels in diabetes by elevating glycosuria.

In this study, we used renal *Slc2a2* knockout mice to identify mechanisms involved in activating a compensatory increase in glucose production in response to glucose loss in urine. We determined whether neural connections between the kidneys and the hypothalamus can explain the compensatory increase in glucose production. In addition, we performed plasma proteomics analyses to identify secreted proteins or endocrine factors that may provide insights into pathways involved in conserving energy and producing endogenous glucose in response to elevated glycosuria in renal *Slc2a2* knockout mice.

## Results and discussion

### Increased hepatic and renal glucose production in renal *Slc2a2* knockout mice

Renal *Slc2a2* knockout mice exhibit normal fasting blood glucose levels despite massive glycosuria (*de Souza Cordeiro et al., 2022*). This observation suggests a compensatory increase in endogenous glucose production, similar to that observed in humans following SGLT2 inhibition (*Solis-Herrera et al., 2020*; *Liu et al., 2012*; *Daniele et al., 2020*). Therefore, we measured hepatic and renal glucose production in renal *Slc2a2* knockout and their littermate control mice. We observed that both tissues contribute to the compensatory increase in glucose production in renal *Slc2a2* knockout mice (*Figure 1a–c*). This was accompanied by a decrease in hepatic glucose 6-phosphate, glucose 1-phosphate, and fructose 6-phosphate (*Figure 1d–f*), suggesting a decline in glucose metabolism in favor of increasing glucose production. A list of measured metabolites is included in *Figure 1—source data 1*.

### Renal *Slc2a2* knockout mice have an activated hypothalamic-pituitary-adrenal (HPA) axis

The HPA axis is a major stress response system involved in enhancing glucose production. Therefore, we measured plasma corticosterone and adrenocorticotropin hormone coupled with gene expression of hypothalamic corticotropic releasing hormone (*Crh*). We observed an increase in all these factors (*Figure 2*) that comprise the HPA axis. These findings suggest that the HPA axis may contribute to the compensatory increase in glucose production in renal *Slc2a2* knockout mice.

### Afferent renal nerves contribute to enhancing glucose production in renal *Slc2a2* knockout mice

To determine the role of afferent renal nerves in triggering a compensatory glucose production in response to glycosuria, we used capsaicin to selectively suppress afferent renal nerve activity in renal *Slc2a2* knockout mice. Afferent denervation was confirmed by measuring renal pelvic CGRP (pg/mg

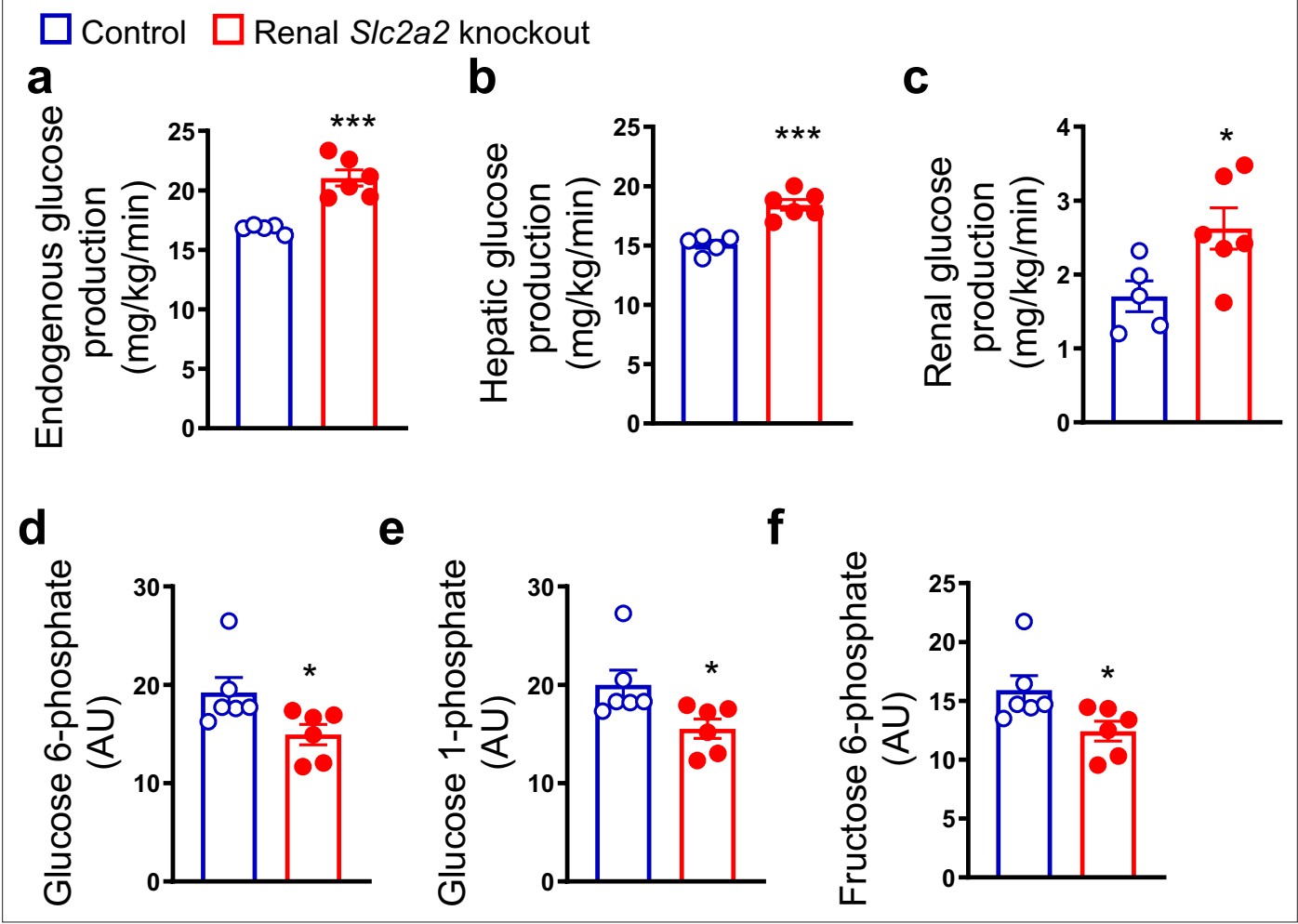

**Figure 1.** Renal *Slc2a2* knockout mice exhibit increased glucose production. In vivo increase in total (**a**), hepatic (**b**), and renal (**c**) glucose production through gluconeogenesis with pyruvate as a substrate in 28-week-old male renal *Slc2a2* knockout mice. Decreased hepatic glucose 6-phosphate (**d**), glucose 1-phosphate (**e**), and fructose 6-phosphate (**f**) in renal *Slc2a2* knockout mice 12 weeks after inducing the *Slc2a2* deficiency. *p<0.05, ***p<0.001, unpaired two-tailed Student's t-test. Data are presented as mean ± SEM.

The online version of this article includes the following source data for figure 1:

**Source data 1.** List of measured metabolites in the liver of renal *Slc2a2* knockout male mice.

protein) at the end of this study (Sham: 86.3±6.4 (control mice), 93.7±8.2 (renal *Slc2a2* knockout mice); Capsaicin: 4.7±0.06 (control mice), 6.4±0.7 (renal *Slc2a2* knockout mice), using ELISA kit, Cayman Chemical, 589001). Mice were allowed to recover for a week before we fasted them to measure their blood glucose levels. Because reinnervation may occur following the denervation, we completed this study within 14 days after the denervation procedure. We observed that fasting and fed (random) blood glucose levels were about 50% decreased in renal *Slc2a2* knockout mice compared to their control littermates (**Figure 3a and b**) after the afferent renal denervation. In addition, the denervation reversed the activation of the HPA axis in renal *Slc2a2* knockout mice (**Figure 3c and d**). Fasting plasma insulin levels were not changed (**Figure 3e**) after the denervation. These observations suggest that renal nerves partly contribute to the compensatory increase in glucose production in renal *Slc2a2* knockout mice. It is important to note that the denervation procedure did not influence baseline blood glucose levels (**Figure 3a and b**) or plasma insulin levels (**Figure 3e**) in the control mice. Similarly, the procedure did not change baseline food intake or body weight in these mice (food intake: 3.7±0.4 vs 4±0.3 g/day; body weight: 26.8±3.6 vs 27.7±5.3 g, sham vs afferent renal denervation).

Although the afferent renal denervation procedure attenuated the compensatory increase in glucose production in renal *Slc2a2* knockout mice, unexpectedly there was no change in their total

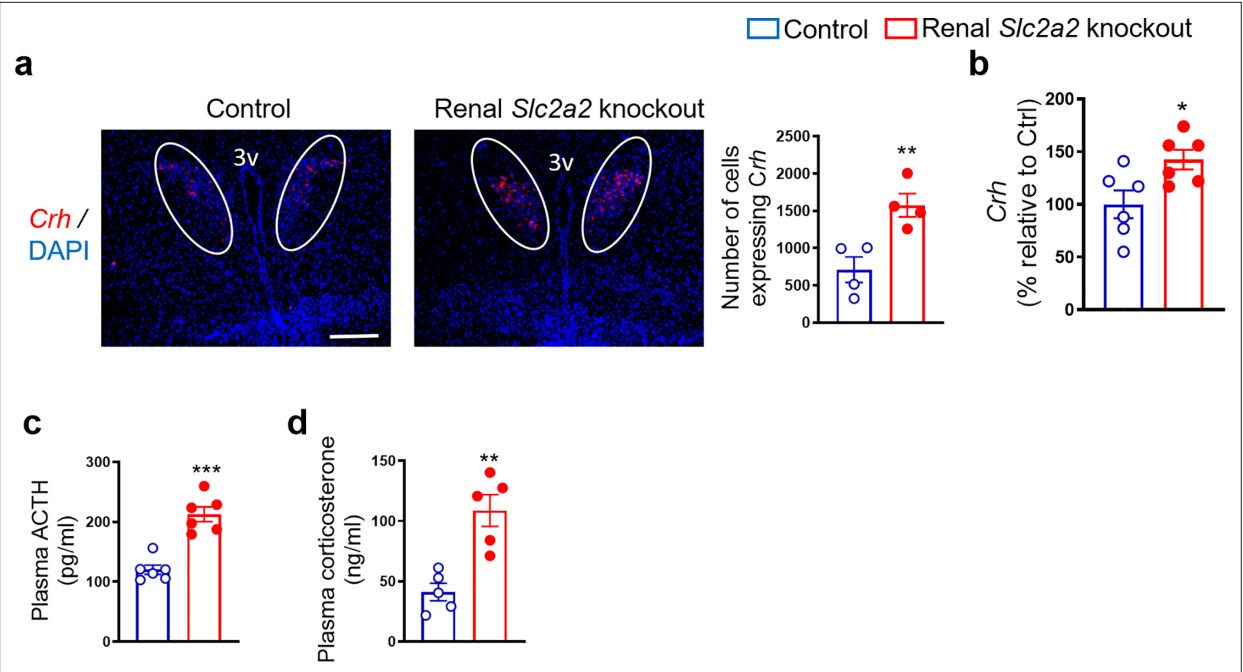

**Figure 2.** Enhanced activity of hypothalamic-pituitary-adrenal axis in renal *Slc2a2* knockout mice. Representative images from fluorescence RNA in situ hybridization showing an increase in expression of corticotropin-releasing hormone (Crh) in the paraventricular nucleus of the hypothalamus (which is identified here using a white oval shape) in 28 weeks old male renal *Slc2a2* knockout mice (**a**). Scale, 100 μm. For the quantification shown next to the images, four sections per mouse and three areas of interest per section were analyzed in four mice. qRT-PCR analysis showing an increase in hypothalamic *Crh* (**b**), data from ELISA demonstrating an increase in plasma adrenocorticotropic hormone (ACTH) (**c**) and corticosterone (**d**) in 12 weeks old male renal *Slc2a2* knockout mice 12 weeks after inducing the *Slc2a2* deficiency. *p<0.05, **p<0.01, ***p<0.001, unpaired two-tailed Student's t-test. Data are presented as mean ± SEM.

or afferent renal nerve activity (*Figure 3—figure supplement 1a and b*). These findings suggest that neuroendocrine mechanisms independent of changes in renal nerve activity may be involved in increasing glucose production in response to elevated glycosuria in renal *Slc2a2* knockout mice. In addition, the *Slc2a2* knockout mice had similar blood pressure, heart rate coupled with similar weights of brown and white adipose tissue, kidney, and liver (*Figure 3—figure supplement 1c–j*). Altogether, the results indicate that afferent renal nerves partly contribute to the compensatory increase in glucose production to make up for glucose loss in renal *Slc2a2* knockout mice. These findings support the data from humans (*Daniele et al., 2020*) that renal nerves may be involved in mediating a compensatory increase in glucose production in response to elevated glycosuria by SGLT2 inhibition. In addition, the findings are aligned with the previous studies in rodents demonstrating the significance of renal nerves in regulating systemic glucose homeostasis (*Chhabra et al., 2017*; *Jiman et al., 2018*).

## Changes in circulating acute phase proteins in renal *Slc2a2* knockout mice

To further investigate underlying mechanisms compensating for glucose loss in *Slc2a2* knockout mice, we measured levels of secreted proteins that may explain the increase in endogenous glucose production. We performed 2D-DIGE followed by proteomic analyses with plasma samples collected from renal *Slc2a2* knockout mice and their littermate controls. We observed that acute-phase proteins like mannose-binding lectin, albumin, haptoglobin, and ferritin were either upregulated or downregulated in renal *Slc2a2* knockout mice (*Figure 4* and *Figure 4—source data 1*). Usually, levels of these proteins are changed in response to threat, infection, and/or injury to activate the body's defense systems (*Gabay and Kushner, 1999*). Similarly, major urinary proteins (MUP) like MUP18, which belong to the lipocalin protein family, were elevated in renal *Slc2a2* knockout mice (*Figure 4* and *Figure 4—source data 1*). These proteins are involved in regulating inter-organ chemical signaling, pheromonal communication, and energy metabolism (*Zhou and Rui, 2010*). We also observed that

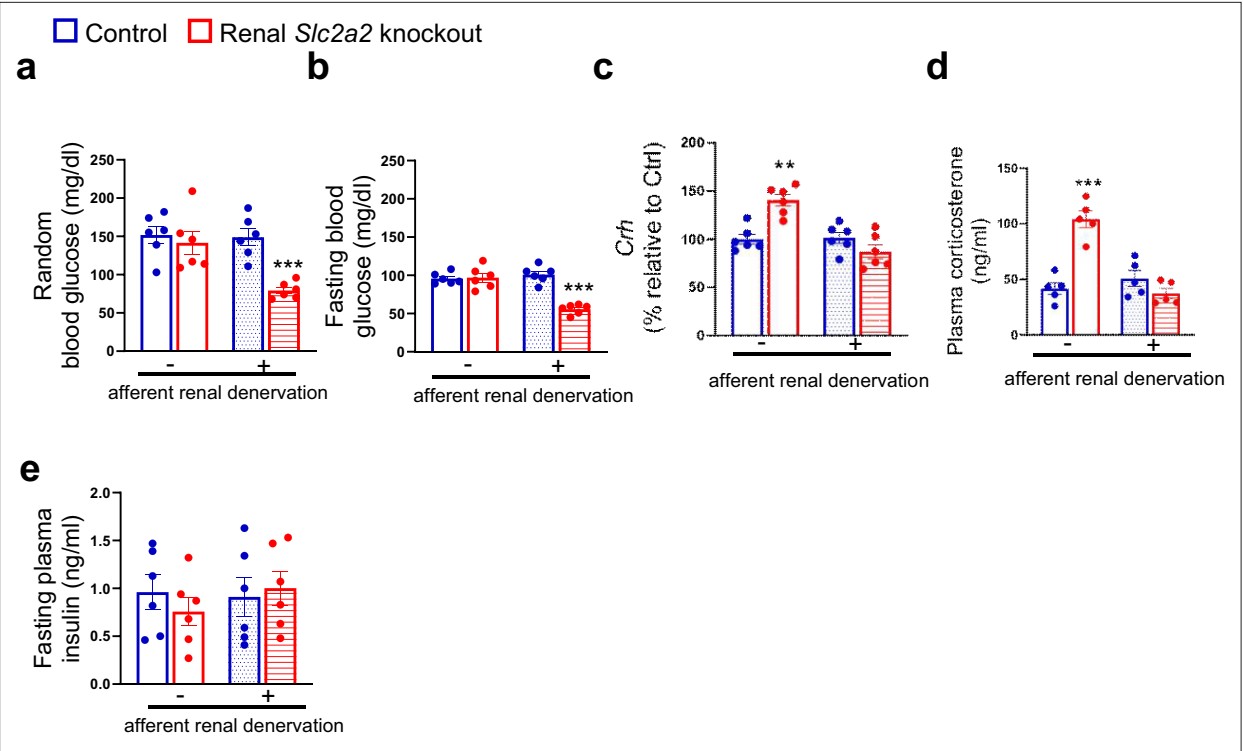

**Figure 3.** Effects of afferent renal denervation on blood glucose levels and hypothalamic-pituitary-adrenal axis in renal *Slc2a2* knockout female mice. Afferent renal denervation decreases fed (random) and fasting (overnight, 6:00 pm – 9:00 am) blood glucose levels (**a, b**), restores expression of hypothalamic corticotropin-releasing hormone (Crh) (**c**) measured using RT-qPCR, and plasma corticosterone (**d**) without affecting plasma insulin levels (**e**) in 30 weeks old female renal *Slc2a2* knockout mice 16 weeks after inducing the *Slc2a2* deficiency. \*\*p0.01, \*\*\*p<0.001, two-way ANOVA followed by a Tukey's post hoc multiple comparison test. Data are presented as mean ± SEM.

The online version of this article includes the following figure supplement(s) for figure 3:

**Figure supplement 1.** No change in renal nerve activities and tissue weights between renal Slc2a2 knockout mice and their littermate controls.

secreted glutathione peroxidase 3 (Gpx3) was the most downregulated protein in plasma in renal *Slc2a2* knockout mice (**Figure 4** and **Figure 4—source data 1**). Gpx3 is predominantly expressed in the kidneys and is an anti-oxidant (**Whitin et al., 2002**). We validated these findings using qRT-PCR. Renal *Gpx3* gene expression was about 50% lower (**Figure 4—figure supplement 1a**) in renal *Slc2a2* knockout mice compared to their littermate controls. Moreover, renal - but not hepatic - *Mup18* gene expression was higher (**Figure 4—figure supplement 1b and c**) in renal *Slc2a2* knockout mice relative to their controls, suggesting that the kidneys were responsible for increasing plasma MUP18 in the knockout mice. Altogether, these results suggest that multiple pathways - including local (renal) stress and general (HPA axis) stress response combined with the acute phase proteins - compensate for glucose loss and defend against a perceived biological threat in renal *Slc2a2* knockout mice.

This study has limitations. Higher urine volume (polyuria) observed in renal *Slc2a2* knockout mice may contribute to an increase in glucose production to some extent, however, that contribution would likely be minor based on previous studies (**Hall et al., 2020**; **Perry et al., 2019**) about the role of diuretics (which increase urine volume) in glucose production. Additional research is necessary to determine the role of the altered secretory proteins – that were identified through the plasma proteomics analyses – in triggering an increase in glucose production in response to elevated glycosuria. It is possible that afferent renal denervation in the present study attenuated only hepatic glucose production through the hypothalamus axis without affecting the compensatory increase in renal (which is the local site of denervation in this study) glucose production. If validated, this would explain why the afferent denervation did not completely block the compensatory glucose production in renal *Slc2a2* knockout mice. Although we didn't observe sex differences in the phenotype related to glucose homeostasis in renal *Slc2a2* knockout mice (**de Souza Cordeiro et al., 2022**), some of the responses to elevated glycosuria reported here may be different between male and female mice.

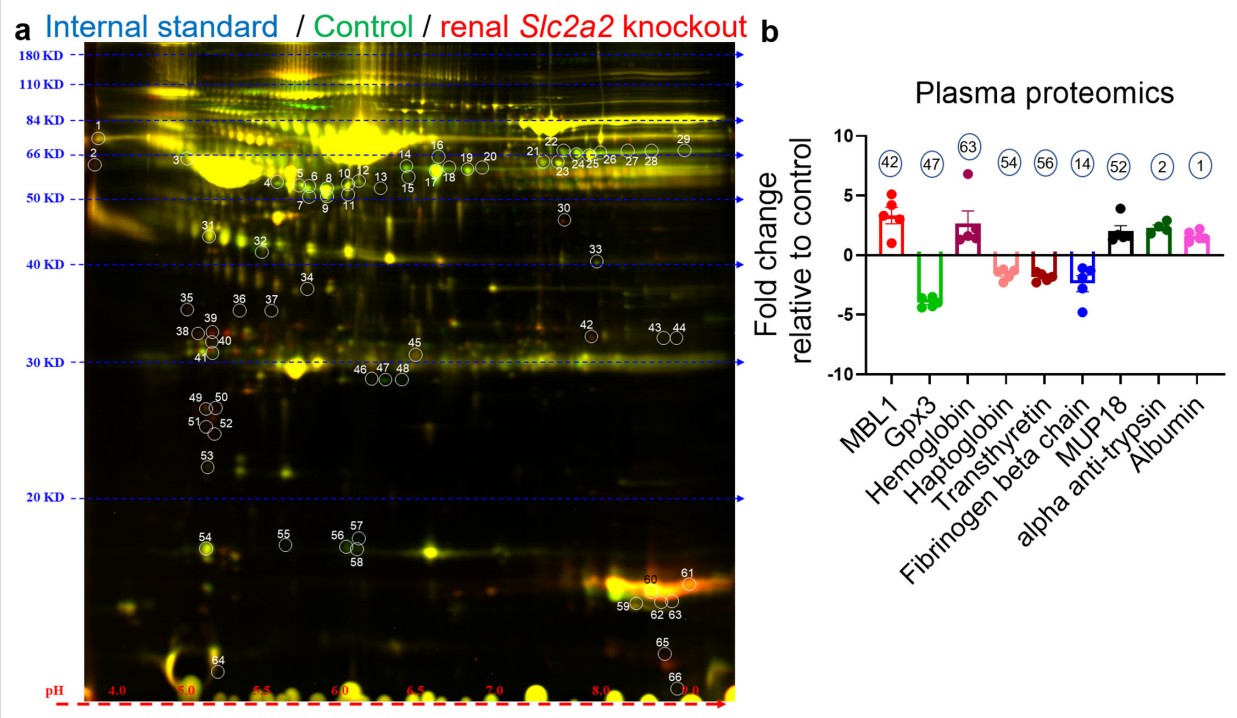

**Figure 4.** Changes in levels of plasma proteins in renal *Slc2a2* knockout male mice 12 weeks after inducing the *Slc2a2* deficiency. Representative image of two-dimensional difference gel electrophoresis with numbered protein spots of interest is shown in (**a**). Internal standard was prepared using equal amounts of protein of each plasma sample as a quality control (Cy2 labeled, pseudo blue), plasma proteins from control group were labeled using Cy3 dye (shown in pseudo green), and plasma proteins from renal *Slc2a2* knockout mice were labeled using Cy5 dye (shown in pseudo red). The identified proteins and their fold change in 28-week-old male renal *Slc2a2* knockout mice compared to the control group are shown in (**b**). The number on each bar graph in (**b**) represents the corresponding protein spot on the gel shown in (**a**). MBL1, mannose binding lectin 1; Gpx3, glutathione peroxidase 3; MUP18, major urinary protein 18.

The online version of this article includes the following source data and figure supplement(s) for figure 4:

**Source data 1.** List of identified proteins in the plasma of renal Slc2a2 knockout male mice.

**Figure supplement 1.** Gene expression analyses in renal Slc2a2 knockout mice.

In summary, we report that a kidney-hypothalamus axis contributes to triggering a compensatory increase in endogenous glucose production in response to elevated glycosuria in mice. Glucose loss in urine appears to be sensed as a biological threat and therefore activates the body's defense systems including some of the acute phase proteins. These findings may explain why SGLT2 inhibitors do not achieve their full potential in lowering blood glucose levels during the treatment of diabetes mellitus.

## Materials and methods
### Mouse husbandry

All animal procedures were approved by the Institutional Animal Care and Use Committee at the University of Rochester (protocol number 2018–010), Yale University (protocol number 2019–20290), or University of Iowa (protocol number 1101549), and were performed according to the US Public Health Service guidelines for the humane care and use of experimental animals. Mice were housed in ventilated cages under controlled temperature (~23 °C) and photoperiod (12 hr light/dark cycle, lights on from 06:00 hr to 18:00 hr or 07:00 hr to 19:00 hr at Yale) conditions with free access to Hydropac water (Lab Products, USA) and regular laboratory chow (5010, LabDiet, USA or Global 2018, Harlan Teklad, USA). Although we included all male or all female mice in a given experiment here, we did not repeat all the experiments in both sexes because our previous study *de Souza Cordeiro et al., 2022* demonstrated that male and female renal *Slc2a2* knockout mice (which were produced in our laboratory as described in *de Souza Cordeiro et al., 2022*) exhibited a similar phenotype compared

to their corresponding control groups in the context of glucose homeostasis. The age and sex of mice used in each experiment in this study are mentioned in the figure legends. After genotyping, mice were randomly assigned to different experimental groups. Renal *Slc2a2* deficiency was induced using tamoxifen (50 mg/kg first dissolved in one part 100% ethanol followed by the addition of nine parts of sesame oil, i.p., T5648, and S3547, Sigma, USA) as described previously in our publication (*de Souza Cordeiro et al., 2022*). *Cdh16*CreERT2 (also known as Ksp-cadherin) or *Slc2a2*loxP/loxP mice were used as littermate controls as described previously in our publication (*de Souza Cordeiro et al., 2022*).

## Glucose production and liver metabolomics

Mice underwent surgery under isoflurane anesthesia to place a catheter in the jugular vein and were allowed to recover for 7 days prior to flux studies as described previously (*Qing et al., 2020*). They were fasted overnight (16 hr) to deplete liver glycogen prior to the study. Mice were administered a 3 X primed-continuous infusion of [3-$^{13}$C] sodium lactate (continuous infusion rate 20 μmol/kg/min) and $^{2}$H$_7$ glucose (continuous infusion rate 2.2 μmol/kg/min) for 120 min, and euthanized with IV Euthasol. Glucose production from pyruvate was directly measured in vivo in the liver and kidney (both freeze-clamped in situ within 30 s of euthanasia) using mass isotopomer distribution analysis as described previously (*Qing et al., 2020*). Briefly, we measured the whole-body ratio of pyruvate carboxylase flux to total glucose production by gas chromatography/mass spectrometry-mass spectrometry. We compared the whole-body gluconeogenesis from phosphoenolpyruvate (PEP) and all substrates upstream of PEP (i.e. pyruvate, lactate, amino acids) to that measured locally in the liver and kidney. As the whole-body fraction of gluconeogenesis from PEP represents an integrated contribution of both liver and kidney to whole-body glucose production, this allowed us to calculate the fractional contribution of the kidney to whole-body glucose production.

For measuring liver metabolites involved in glycolysis, the flash-frozen liver was homogenized to extract metabolites in 80% methanol, which was evaporated under liquid nitrogen followed by resuspension of metabolites in 50% methanol. Liquid Chromatography-Tandem Mass Spectrometry (LC-MS/MS) analysis was performed with reverse-phase LC on a Synergi Fusion-RP column (Phenomenex, Torrence CA) at 35 °C. Samples were analyzed by single reaction monitoring on a Thermo Quantum triple-quadrupole mass spectrometer. Metabolites were identified against a library of validated standards based on retention time, intact mass, collision energy, and fragment masses. Data were collected and analyzed using Thermo XCaliber 4.0 software. Post-hoc data was analyzed using Metaboanalyst 5.0. Data were median-normalized (normalized to the median peak area per sample, which correlated well with the total protein concentration). 73 metabolites were measured, out of which nine were significantly (p<0.05) different (1.5 fold-change thresholds, *Figure 1—source data 1*).

## Measurement of genes and hormones involved in regulating hypothalamic-pituitary-adrenal axis

We measured the levels of *Crh* mRNA in the mouse paraventricular hypothalamus using RNA fluorescence in situ hybridization (*Crh* probe, 316091, ACD) according to the manufacturer's instructions. DAPI was used to stain the nucleus and verify the regions of interest in the mouse brain. Images were captured using Keyence fluorescence microscope BZ-X800. To further quantify the number of cells expressing *Crh* or DAPI, we used CellProfiler 4.4.1 (https://cellprofiler.org/releases).

We used the following primers to measure gene expression using RT-qPCR: *Crh*: 5'-GGAATCTC AACAGAAGTCCCGC-3' and 5'-CTGCAGCAACACGCGGAAAAAG-3', *Hrpt*: 5'-AACAAAGTCTGG CCTGTATCC-3' and 5'-CCCCAAAATGGTTAAGGTTGC-3'. All primers were used at a final concentration of 500 nmol/l. The relative quantity of each mRNA was calculated from standard curves and normalized to the internal control Hprt, and then normalized to the mean of corresponding controls.

To minimize a natural stress response, mice were acclimated to handling procedures at least three days before collecting their blood. For measuring adrenocorticotropic hormone and corticosterone, we collected mouse tail blood at 10:00 hr on the day of measurements by nicking the tip of the tail with a sterile razor blade and then using heparinized capillary tubes (22–362566, Thermo Fisher Scientific). For measuring plasma insulin, mice were fasted for 6 hr (08:00–14:00 hr) before collecting their tail blood. The blood was centrifuged for 10 min at 2000 × g and 4 °C. Plasma levels of adrenocorticotropic hormone (ab263880, Abcam), corticosterone (80556, Crystal Chem), and insulin (90080, Crystal

Chem) were measured using ELISA per the manufacturers' instructions. The plasma was diluted 5 x before performing these assays and the results were adjusted according to the dilution factor.

### Ablation of afferent renal nerve

After anesthetizing mice using 1–5% isoflurane, a dorsal midline incision was made to access both the kidneys. We selectively ablated afferent renal nerves using capsaicin. Gauze, soaked in capsaicin (35 mM in 10% ethanol and 90% saline solution), was wrapped around the renal blood vessels for 15 min. Other surrounding tissues were covered with parafilm to avoid their exposure to capsaicin. After the surgery, muscle layers and skin were closed with nylon sutures. This procedure ablates afferent renal nerves without affecting the blood vessels or efferent renal nerves (*Foss et al., 2015*). Control mice underwent the same procedure without capsaicin (sham surgery using 10% ethanol and 90% saline solution). The mice were allowed to recover for a week before determining the effects of the afferent denervation on blood glucose and other parameters.

### Measurement of renal nerve activity, blood pressure, and heart rate

The mouse renal nerve activity was measured using multifiber recording as described previously (*Morgan and Rahmouni, 2010*). After anesthesia, the nerve innervating the left kidney was identified, dissected free, and placed on a bipolar 36-gauge platinum-iridium electrode (A-M Systems; Carlsborg, WA). The electrode was connected to a high-impedance probe (HIP-511; Grass Instruments Co., Quincy, MA), and the nerve signal was amplified $10^5$ times with a Grass P5 AC pre-amplifier and filtered at low and high-frequency cutoffs of 100 Hz and 1000 Hz, respectively. This nerve signal was directed to a speaker system and to an oscilloscope (54501 A, Hewlett–Packard Co., Palo Alto, CA) for auditory and visual monitoring of the nerve activity. The signal was then directed to a resetting voltage integrator (B600C, University of Iowa Bioengineering) that sums the total voltage output in units of 1 V × s before resetting to zero and counting the number of spikes per second. The final neurograms were continuously routed to a MacLab analogue–digital converter (8 S, AD Instruments Castle Hill, New South Wales, Australia) for permanent recording and data analysis on a Macintosh computer. Renal sympathetic nerve activity was measured for 30 min. To measure the afferent renal nerve activity, the nerve was cut distally and the signal was recorded for 30 min. Nerve activity was corrected for post-mortem background activity to eliminate background electrical noise in the measurements. At the end of the SNA recording, each mouse was euthanized with an overdose of anesthetic. Any remaining nerve activity after death was considered as background noise and subtracted from the SNA measurements.

In addition to the nerve activity, arterial pressure and heart rate were measured using a tapered MRE-040 tubing inserted into the left carotid artery and the other end connected to a transducer (BP-100; iWorks System, Inc, Dover, NH) that led to an ETH-250 Bridge/Bio Amplifier (CB Sciences; Milford, MA). Core body temperature of the mouse was measured using a rectal probe (YSI 4000 A Precision Thermometer; Yellow Springs, OH) and maintained constant at 37.5 °C using a custom-made heated surgical platform.

### Two-dimensional difference gel electrophoresis (2D-DIGE)

Plasma proteins from control and renal *Slc2a2* knockout mice were labeled with Cy3 and Cy5 dyes, respectively. We also used an internal standard (labeled with Cy2 dye) containing equal amounts of protein of each plasma sample as a quality control. Using 2D gel electrophoresis, we separated plasma proteins (30 µg in 2D sample buffer) on a gel by both isoelectric point and molecular weight. We then imaged the gel using a Typhoon scanner. DeCyder software was used to identify protein spots of interest (downregulated or upregulated by at least 25%), which were automatically picked from the 2D gel with Ettan Spot Picker (Amersham BioSciences). The proteins were identified by MALDI TOF/ TOF mass spectrometry (AB SCIEX TOF/TOF 5800 System) using the Swiss-Prot database.

### Statistics

Data are shown as mean ± SEM. Results were analyzed by two-tailed Student's unpaired t-test or two-way ANOVA followed by a Tukey's post hoc multiple comparison test when appropriate. All analyses were performed using Prism version 8.0.1 (GraphPad, USA) and differences were considered statistically significant at $p < 0.05$.

## Acknowledgements

We thank V Kaye Thomas, and Julie Zhang, URMC Center for Advanced Light Microscopy and Nanoscopy, for help with microscopy; and Applied Biomics for proteomics analyses. National Institutes of Health grant CA258261 to RJP. National Institutes of Health grant HL071158 PSB. National Institutes of Health grant DK124619 to KHC. Startup funds, Department of Medicine, University of Rochester, NY to KHC.

## Additional information

### Funding

| Funder | Grant reference number | Author |
| --- | --- | --- |
| National Institutes of Health | DK124619 | Kavaljit H Chhabra |
| National Institutes of Health | CA258261 | Rachel J Perry |
| National Institutes of Health | HL071158 | Paul S Brookes |

The funders had no role in study design, data collection and interpretation, or the decision to submit the work for publication.

### Author contributions

Tumininu S Faniyan, Data curation, Investigation, Methodology, Writing – review and editing; Xinyi Zhang, Donald A Morgan, Siresha Bathina, Data curation, Formal analysis, Investigation, Methodology, Writing – review and editing; Jorge Robles, Data curation, Formal analysis, Investigation, Writing – review and editing; Paul S Brookes, Resources, Data curation, Formal analysis, Supervision, Funding acquisition, Validation, Investigation, Methodology, Writing – review and editing; Kamal Rahmouni, Conceptualization, Resources, Data curation, Formal analysis, Supervision, Funding acquisition, Validation, Investigation, Methodology, Writing – review and editing; Rachel J Perry, Conceptualization, Resources, Data curation, Formal analysis, Supervision, Funding acquisition, Validation, Investigation, Methodology, Writing - original draft, Project administration, Writing – review and editing; Kavaljit H Chhabra, Conceptualization, Resources, Data curation, Formal analysis, Supervision, Funding acquisition, Validation, Investigation, Visualization, Methodology, Writing - original draft, Project administration, Writing – review and editing

### Author ORCIDs

Paul S Brookes ⓘ https://orcid.org/0000-0002-8639-8413
Rachel J Perry ⓘ https://orcid.org/0000-0003-0748-8064
Kavaljit H Chhabra ⓘ https://orcid.org/0000-0001-6378-3645

### Ethics

This study was performed in strict accordance with the recommendations in the Guide for the Care and Use of Laboratory Animals of the National Institutes of Health. All of the animals were handled according to approved institutional animal care and use committee (IACUC) protocols at the University of Rochester (protocol number 2018-010), Yale University (protocol number 2019-20290), and the University of Iowa (protocol number 1101549).

Reviewer #1 (Public Review): https://doi.org/10.7554/eLife.91540.4.sa1
Author response https://doi.org/10.7554/eLife.91540.4.sa2

### Data availability

All data are available in the main text or the source data files. The reagents and mouse model used in this study are available via material transfer agreement addressed to the corresponding author.

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
