## [Editor Report · eLife assessment]

The study presents **valuable** findings on compensatory mechanisms in response to glycosuria. The evidence supporting the claims is **solid**, although a causal relationship is somewhat uncertain and the addition of a more clinically relevant model would have strengthened the findings. The work will be of interest to diabetes investigators.

---

## [Referee Report · Reviewer #1 (Public Review)]

Summary:

In this study, Faniyan and colleagues build on their recent finding that renal Glut2 knockout mice display normal fasting blood glucose levels despite massive glucosuria. Renal Glut2 knockout mice were found to exhibit increased endogenous glucose production along with decreased hepatic metabolites associated with glucose metabolism. Crh mRNA levels were higher in the hypothalamus while circulating ACTH and corticosterone was elevated in this model. While these mice were able to maintain normal fasting glucose levels, ablating afferent renal signals to the brain caused low fasting blood glucose levels. In addition, the higher CRH and higher corticosterone levels of the knockout mice were lost following this denervation. Finally, acute phase proteins were altered, plasma Gpx3 was lower, and major urinary protein MUP18 and its gene expression were higher in renal Glut2 knockout mice. Overall, the main conclusion that afferent signaling from the kidney is required for renal glut2 dependent increases in endogenous glucose production is well supported by these findings.

Strengths:

An important strength of the paper is the novelty of the identification of kidney to brain communication as being important for glucose homeostasis. Previous studies had focused on other functions of the kidney modulated by or modulating brain function. This work is likely to promote interest in CNS pathways that respond to afferent renal signals and the response of the HPA axis to glucosuria. Additional strengths of this paper stem from the use of incisive techniques. Specifically, the authors use isotope enabled measurement of endogenous glucose production by GC-MS/MS, capsaicin ablation of afferent renal nerves, and multifiber recording from the renal nerve. The authors also paid excellent attention to rigor in the design and performance of these studies. For example, they used appropriate surgical controls, confirmed denervation through renal pelvic CGRP measurement, and avoided the confounding effects of nerve regrowth over time. These factors strengthen confidence in their results. Finally, humans with glucose transporter mutations and those being treated with SGLT2 inhibitors show a compensatory increase in endogenous glucose production. Therefore, this study strengthens the case for using renal Glut2 knockout mice as a model for understanding the physiology of these patients.

Comments on latest version:

My concerns have been addressed.

---

## [Author Response]

The following is the authors’ response to the previous reviews.

**Reviewer #1 (Public Review):**
Summary:In this study, Faniyan and colleagues build on their recent finding that renal Glut2 knockout mice display normal fasting blood glucose levels despite massive glucosuria. Renal Glut2 knockout mice were found to exhibit increased endogenous glucose production along with decreased hepatic metabolites associated with glucose metabolism. Crh mRNA levels were higher in the hypothalamus while circulating ACTH and corticosterone was elevated in this model. While these mice were able to maintain normal fasting glucose levels, ablating afferent renal signals to the brain caused low fasting blood glucose levels. In addition, the higher CRH and higher corticosterone levels of the knockout mice were lost following this denervation. Finally, acute phase proteins were altered, plasma Gpx3 was lower, and major urinary protein MUP18 and its gene expression were higher in renal Glut2 knockout mice. Overall, the main conclusion that afferent signaling from the kidney is required for renal glut2 dependent increases in endogenous glucose production is well supported by these findings.Strengths:An important strength of the paper is the novelty of the identification of kidney to brain communication as being important for glucose homeostasis. Previous studies had focused on other functions of the kidney modulated by or modulating brain function. This work is likely to promote interest in CNS pathways that respond to afferent renal signals and the response of the HPA axis to glucosuria. Additional strengths of this paper stem from the use of incisive techniques. Specifically, the authors use isotope enabled measurement of endogenous glucose production by GC-MS/MS, capsaicin ablation of afferent renal nerves, and multifiber recording from the renal nerve. The authors also paid excellent attention to rigor in the design and performance of these studies. For example, they used appropriate surgical controls, confirmed denervation through renal pelvic CGRP measurement, and avoided the confounding effects of nerve regrowth over time. These factors strengthen confidence in their results. Finally, humans with glucose transporter mutations and those being treated with SGLT2 inhibitors show a compensatory increase in endogenous glucose production. Therefore, this study strengthens the case for using renal Glut2 knockout mice as a model for understanding the physiology of these patients.Weaknesses:A few weaknesses exist. Most concerns relate to the interpretation of this study's findings. The authors state that loss of glucose in urine is sensed as a biological threat based on the HPA axis activation seen in this mouse model. This interpretation is understandable but speculative. Importantly, whether stress hormones mediate the increase in endogenous glucose production in this model and in humans with altered glucose transporter function remains to be demonstrated conclusively. For example, the paper found several other circulating and local factors that could be causal. This model is also unable to shed light on how elevated stress hormones might interact with insulin resistance, which is known to increase endogenous glucose production. That issue is of substantial clinical relevance for patients with T2D and metabolic disease. Finally, how these findings can contribute to improving the efficiency of drugs like SGLT2 inhibitors remains to be seen.

- We agree with the reviewer’s overall assessment of this manuscript.

- Confirming the contribution of each secreted protein shown in Fig. 4, whose levels were changed between the two groups of mice, toward causing a compensatory increase in glucose production in response to elevated glycosuria is beyond the scope of this manuscript.

**Reviewer #2 (Public Review):**
Summary:The authors previously generated renal Glut2 knockout mice, which have high levels of glycosuria but normal fasting glucose. They use this as an opportunity to investigate how compensatory mechanisms are engaged in response to glycosuria. They show that renal and hepatic glucose production, but not metabolism, is elevated in renal Glut2 male mice. They show that renal Glut2 male mice have elevated Crh mRNA in the hypothalamus, and elevated plasma levels of ACTH and corticosterone. They also show that temporary denervation of renal nerves leads to a decrease in fasting and fed blood glucose levels in female renal Glut2 mice, but not control mice. Finally, they perform plasma proteomics in male mice to identify plasma proteins that are changed (up or down) between the knockouts and controls.Strengths:The question that is trying to be addressed is clinically important: enhancing glycosuria is a current treatment for diabetes, but is limited in efficacy because of compensatory increases in glucose production.Weaknesses:(1) Although I appreciate that the initial characterization of the mice in another publication showed that both males and females have glycosuria, this does not mean that both sexes have the same mechanisms giving rise to glycosuria. There are many examples of sex differences in HPA activation in response to threat, for example. There is an unfounded assumption here that males and females have the same underlying mechanisms of glycosuria that undermines the significance of the findings.

- We agree with the reviewer that although we didn’t observe sex differences in renal Glut2 KO mice in the context of glucose homeostasis, their response (or mechanisms) to elevated glycosuria in enhancing compensatory glucose production may be different between the sexes. Therefore, we have included this limitation in discussion section.

(2) The authors state that they induced the Glut2 knockout with taxomifen as in their previous publication. The methods of that publication indicate that all experiments were completed within 14 days of inducing the Glut2 knockout. This means that the last dose of tamoxifen was delivered 14 days prior to the experimental endpoint of each experiment. This seems like an important experimental constraint that should be discussed in this manuscript. Is the glycosuria that follows Glut2 knockout only a temporary change? If so, then the long-term change in glycosuria that follows SGLT2 inhibition in humans might not be best modelled by this knockout. Please specify when the surgeries to implant a jugular catheter or ablate the renal nerves performed relative to the Glut2 knockout in the Methods.

- The reviewer’s statement ‘The methods of that publication indicate that all experiments were completed within 14 days of inducing the Glut2 knockout’ is incorrect. In the referred publication, we had explicitly mentioned in methods, ‘All of the experiments, except those using a diet-induced obesity mouse model or noted otherwise, were completed within 14 days of inducing the Glut2 deficiency.’ Please see figures 5h-l and 6 in the cited publication, which demonstrate that all the experiments were not completed within 14 days of inducing renal Glut2 deficiency. Per the reviewer’s advice, in the present manuscript we have include the timeline (which in some cases is 4 months beyond inducing glycosuria) in all the figure legends. In addition, for a separate project (which is unpublished) we have measured glycosuria up to 1 year after inducing renal Glut2 deficiency. Therefore, the glycosuria observed in the renal Glut2 KO mice is not temporary.

(3) I am still unclear what group was used for controls. Are these wild-type mice who receive tamoxifen? Are they KspCadCreERT2;Glut2loxP/loxP mice who do not receive tamoxifen? This is important and needs to be specified.

- In our previous response to the reviewer, we had already mentioned which control group was used in this study. Please see our response to the second reviewer’s point 3. As mentioned to the reviewer, we had used Glut2loxp/loxp mice as the control group, which is also described multiple times in the figure legends of our previous paper that reported the phenotype of renal Glut2 KO mice. Per the reviewer’s advice, we have provided the information again in a revised version of this manuscript.

(4) The authors should report some additional control measures for the renal denervation that could also impact blood glucose and perhaps some of their other measures. The control measures, which one would like to see unimpacted by renal denervation, include body weights, food consumption and water intake, and glycosuria itself.

- Please also see fig. 3 in the present manuscript that demonstrates renal afferent denervation doesn’t influence baseline blood glucose or plasma insulin levels. We have now also mentioned in the text that the denervation doesn’t affect food intake or bodyweight.

(5) The graphical abstract shows a link between the hypothalamus and the liver that is completely unsupported by any of the current findings. That arrow should be removed.

- Because we observed an increase in hepatic glucose production in renal Glut2 KO mice (Fig. 1) - which was reduced by 50% after selective afferent renal denervation (Fig. 3) - in the graphical abstract we are suggesting a neural connection between the kidney-brain-liver or an endocrine factor(s) to account for these changes in blood glucose levels as also described in the discussion section. We can include a question mark ‘?’ in the graphical abstract to show that further studies are need to validate these proposed mechanisms; however, we cannot just remove the arrow as advised by the reviewer.

(6) Though the authors have toned down their language implying a causal link between the HPA measures and compensatory elevation of blood glucose in the face of glycosuria, the title still implies this causal link. It is still the case that their data do not support causation. There are many potential ways to establish a causal link but those experiments are not performed here. The renal afferents are correlated with Crh content of the PVN, but nothing has been done to show that the Crh content is important for elevating blood glucose. In light of this, the title should be toned down. Perhaps something like "Renal nerves maintain blood glucose production and elevated HPA activity in response to glycosuria". The link between HPA and glucose is not shown in this paper.

- We request the reviewer to take a look at figure 1, showing an increase in glucose production in renal Glut2 KO mice and figure 3, which demonstrates that an afferent renal denervation reduces blood glucose levels by 50%. The afferent renal denervation (ablation of afferent renal nerves) does reduce blood glucose levels in renal Glut2 KO mice. Therefore, the use of the word ‘promote’ in the title is accurate and appropriate to reflect the role of the afferent renal nerves in contributing to about 50% increase in blood glucose levels in renal Glut2 KO mice.

- Regarding the reviewer's comment on changes in Crh gene expression, please look at figure 3. Ablation of renal afferent nerves decreases hypothalamic Crh gene expression and other mediators of the HPA axis by 50%. Therefore, the afferent renal nerves do contribute to regulating blood glucose levels, at least in part, by the HPA axis (which is widely known to change blood glucose levels). The use of words such as ‘required’ or ‘necessary’ in the title may have indicated causal role or could have been misleading here; therefore, we have purposely used ‘promote’ in the title to accurately reflect the findings of this study.

**Recommendations for the authors:**

**Reviewer #1 (Recommendations For The Authors):**
I have only minor text corrections to add:- line 223 "A list"- line 253 "independent"- line 271 "the body's"- line 304 "do not"

Yes, we have corrected these errors in a revised version of this manuscript.

**Reviewer #2 (Recommendations For The Authors):**
(1) Please report the dilutions used, if any, for the ELISAs. If the samples were run neat, please report this. Many manufacturer's instructions say that the user must determine the correct dilution to use for the samples collected. Also, sometimes when small blood volumes are collected, samples must be diluted to achieve the minimum volume collected for the assay. It is not sufficient to indicate that a reader refers to the manufacturer's instructions.

- Per the reviewer’s advice, we have included the dilutions used for each assay in the methods.